# In Vitro Shoot Multiplication and Regeneration of the Recalcitrant Rocket (*Eruca sativa* Mill.) Variety Domaća Rukola

Nevena Banjac *[ID], Dijana Krstić-Milošević [ID], Tatjana Mijalković, Mirjana Petrović [†], Tatjana Ćosić [ID], Mariana Stanišić and Branka Vinterhalter [ID]

Department of Plant Physiology, Institute for Biological Research "Siniša Stanković"—National Institute of the Republic of Serbia, University of Belgrade, Bulevar despota Stefana 142, 11000 Belgrade, Serbia; dijana@ibiss.bg.ac.rs (D.K.-M.); mijalkovic.t24@gmail.com (T.M.); mpetrovic@torlak.rs (M.P.); tatjana@ibiss.bg.ac.rs (T.Ć.); mariana.stanisic@ibiss.bg.ac.rs (M.S.); horvat@ibiss.bg.ac.rs (B.V.)
* Correspondence: mitic.nevena@ibiss.bg.ac.rs
† Current address: Institute of Virology, Vaccines and Sera "Torlak", Vojvode Stepe St. 458, 11152 Belgrade, Serbia.

**Abstract:** Eruca sativa is known in traditional medicine for its therapeutic effects, while young plants are used as a salad or green food. Recently, the consumption of rocket has increased considerably, so it has become very important for breeders. Plant tissue culture provides a platform to overcome the problems in improving this species. In the present study, an efficient protocol for in vitro shoot regeneration and propagation of recalcitrant rocket variety Domaća rukola was studied. Murashige and Skoog (MS, 1962) medium containing 0.1 mg L$^{-1}$ BA and frequent subculture over a period of three weeks proved to be optimal for shoot multiplication with a multiplication index of over 3 and only 8.72% of hyperhydrated shoots without necrosis. Different concentrations of 2,4-D, BA, or TDZ in combination with NAA, with or without the presence of AgNO$_3$, were tested for de novo shoot organogenesis (DNSO) from seedling explants. The hypocotyl explants cultured on MS with a combination of TDZ1.0+NAA0.1+AgNO$_3$ 5.0 mg L$^{-1}$ regenerated viable shoots with the highest rate (25.38%) and an average number of 2.18 shoots per regenerating explant. Somatic embryogenesis from immature zygotic embryos proved to be the best way to regenerate a recalcitrant rocket cultivar. The highest embryogenic efficiency was achieved in explants cultured on MS medium containing 1.0 mg L$^{-1}$ 2,4-D with a frequency of 76.64% and 5.13 mean number of regenerated somatic embryos per explant, which were further converted into normal plants. Additionally, in vitro-produced rocket shoots could serve as a possible promising source for the production of flavonoid kaempferol with proven antioxidant properties.

**Keywords:** organogenesis; somatic embryogenesis; cytokinins; rooting; kaempferol

---





## 1. Introduction

Rocket (*Eruca sativa* Mill., Brassicaceae) is an annual herbaceous plant with a characteristic odor. It originates from the Mediterranean region, from where it has spread to North America, Asia, Europe, and Africa [1].

Although it is mainly used in the diet as a salad because of its spicy taste, the use of rocket is multiple. It is a source of edible oils and proteins [2,3], so the consumption of rocket has significantly increased recently. Its leaves are also used in herbal medicine [4]. Rocket has shown antioxidant potential [5] and has a protective effect against mercuric chloride (HgCl$_2$)-induced injuries [6–10]. Rocket seeds produce 4-methylthiobutyl glucosinolate (MTG) [11] and long-chain erucinic fatty acid (C22) [12]. MTG hydrolysis products, isothiocyanates, have anticancer activity because they inhibit mitosis and promote apoptosis of tumor cells, increasing immunity in humans [13]. Rocket also contains flavonoids, alkaloids, and terpenoids that have antimicrobial activity [14] and is an available resource for the production of low molecular weight antioxidants. Phenolics and flavonoids in the

---

aqueous extracts of *E. sativa* have been shown to play a major role in the detoxification of free radicals [10,15].

Consequently, the rocket is considered an available genetic pool for the improvement of other members of the Brassicaceae family, as well as a model system for research on long-chain fatty acids metabolism, biosynthesis of glucosinolates, and stress resistance [16]. However, rocket cultivation is affected by various plant diseases and pests [17,18]. Effects can range from minor damage to complete crop loss, depending on the pathogen involved. Insect pests, including plant aphides and flea beetles, are a common threat, especially to plants grown in indoor conditions such as greenhouses and plastic shelters [19,20]. They can cause significant damage to the leaves, which represent the marketable product of rocket, making them unprofitable. In addition, flower damage can affect the production of seeds, which are the source of Eruca oil and important for further plant production.

Unfortunately, the conventional rocket breeding program is limited by the biennial plant nature and the need for strong barriers in order to prevent unwanted hybridization with other rocket lines or related *Brassica* species. Plant tissue culture offers a platform to overcome existing barriers to improve the quality, as well as disease and stress resistance of this important crop. A number of protocols for in vitro propagation of rocket have reported de novo shoot organogenesis (DNSO), using cytokinins, mainly 6-benzyladenine (BA) and kinetin (KIN), applied alone or in combination with auxins on hypocotyl, cotyledon, stem, and leaf explants [21–26]. However, most of these protocols were complicated, while the rocket response to exogenous growth regulators varied significantly and depended mainly on the variety.

In addition to DNSO, the production of somatic embryos (SEs) can be very important for breeding purposes. Unfortunately, somatic embryogenesis has rarely been reported in rocket, and so far, only a few reports are available. Ashloowalia [21] first reported somatic embryogenesis in the culture of zygotic embryos of *E. vesicaria* ssp. *sativa*, while Parkash et al. [22] and Sikdar et al. [27] obtained somatic embryogenesis in the culture of mesophyll protoplasts of *E. sativa*. More recently, Zhang et al. [16] and Chen et al. [12] investigated the SE regeneration from cotyledon and hypocotyl explants of *E. sativa*. However, despite this, no report on a universal and effective protocol for shoot regeneration in *Eruca* is available. In an attempt to establish a regenerative protocol for the rocket variety Domaća rukola, previously established protocols applied to this variety were unsatisfactory due to weak or absent regenerative response.

This indicates that in vitro regeneration of rocket is not yet routine and, therefore, implies additional optimization regarding the choice of the type of starting explants, as well as the combination and concentration of plant growth regulators (PGRs).

Since the regeneration response was mostly influenced by genotype and considering that significant variations among the same species, or even cultivar, are feasible, further efforts must be made to optimize in vitro rocket regeneration, especially in the development of the procedure for SE induction.

Therefore, in the present work, a comprehensive study was conducted to achieve an efficient regeneration protocol for in vitro rocket production. In vitro propagation of variety Domaća rukola was examined using three different pathways, including shoot proliferation from seedling epicotyls, de novo shoot organogenesis from seedling explants, and somatic embryogenesis from immature zygotic embryos. The use of $AgNO_3$ was also evaluated in order to improve regeneration efficiency. Since immature embryos are an available source for the production of shoots of newly established lines for the purposes of conservation and breeding, the conditions for somatic embryogenesis from immature embryos are presented. In addition, HPLC analysis of secondary metabolites was made to compare their qualitative and quantitative content between the in vitro and control seed-derived plants.

## 2. Materials and Methods

### 2.1. Plant Material and Culture Conditions

Mature seeds of rocket variety Domaća rukola and siliques with immature seeds, harvested from plants grown in a greenhouse at the Institute of Biological Research "Siniša Stanković" in Belgrade, Serbia, were surface-disinfected with 70% ethanol for 1 min, then soaked in 20% (*w/v*) commercial bleach (NaOCl) solution with a drop of Fairy® detergent for 25 min, and rinsed 3–4 times with sterile distilled water. Disinfected mature seeds were germinated on $\frac{1}{2}$ Murashige and Skoog (MS) [28] medium with 0.64% (*w/v*) agar (Torlak, Belgrade, Serbia) and 2% sucrose (Torlak, Belgrade, Serbia), filled in 90 × 15 mm Petri dishes.

Basal MS medium containing MS mineral salts, LS vitamins [29], 0.64% (*w/v*) agar (Torlak, Belgrade, Serbia), 2% sucrose (Torlak, Belgrade, Serbia), and 100 mg L$^{-1}$ myoinositol (Thermo Fisher, Bremen, Germany) was used for the tissue culture experiments. The pH of all media was adjusted to 5.8.

Cultures were grown at 25 ± 2 °C in a controlled-environment room illuminated with cool-white Phillips fluorescent lamps, providing 35–45 μmol m$^{-2}$ s$^{-1}$ under a 16-h photoperiod (long day).

### 2.2. Shoot Culture Establishment, Multiplication, and Rooting

For shoot culture establishment and multiplication, epicotyls were cut from five-day-old seedlings and transferred to Erlenmeyer flasks containing 40 mL of basal MS medium supplemented with 0.1 mg L$^{-1}$ and 0.2 mg L$^{-1}$ of either BA (6-benzyladenine, Sigma–Aldrich Chemie GmbH, Taufkirchen, Germany or KIN (kinetin, 6-phurphurylaminopurine, Sigma–Aldrich Chemie GmbH, Taufkirchen, Germany. The number of multiplied shoots and their viability were recorded after three weeks of culture. The multiplication index was calculated as (1 shoot explant + the number of newly formed shoots (length ≥ 2 mm))/total number of shoot explants per treatment. Proliferated shoots with a length of 1.5–3.0 cm were separated from shoot clusters and individually transferred onto a medium with $\frac{1}{2}$MS mineral salts and IBA (indole-3-butyric acid, Sigma–Aldrich, Chemie GmbH, Taufkirchen, Germany) at increasing concentrations of 0, 0.1, 0.2, 0.5, and 1.0 mg L$^{-1}$. Rooting percentage, number of roots, and length of the longest root were scored after 30 days of culture.

### 2.3. Extract Preparation and HPLC Analysis of Flavonoids in Multiplied Shoots

Air-dried samples of control seed-derived rocket plants grown in a greenhouse and plants multiplied in vitro in medium containing 0.1 and 0.2 mg L$^{-1}$ BAP or KIN were ground in a laboratory mill and extracted with methanol for 20 min in an ultrasonic bath. After sonication, extraction was continued by maceration for 24 h in the dark at room temperature. The extracts were filtered, and the solvent was evaporated to dryness in a rotary vacuum evaporator at 50 °C. The dry extracts were stored at 4 °C until used for HPLC analysis.

For identification of flavonoid aglycones, the dry extracts (10 mg) were dissolved in methanol (1 mL) and hydrolyzed with 4 M HCl in a water bath at 85 °C for 30 min according to the method described by Martínez-Sánchez et al. [30]. Separation, identification, and quantification of flavonoids were performed on an Agilent 1100 series HPLC instrument (Agilent, Waldbronn, Germany) with a DAD detector on a Zorbax SB-C18 (Agilent) reversed-phase analytical column (150 mm × 4.6 mm i.d., 5 μm particle size). The temperature was set at 30 °C. The mobile phase consisted of solvent A (0.1%, *v/v* solution of orthophosphoric acid in water) and solvent B (acetonitrile), and the elution gradient was as follows: 90–80% A 0–10 min; 80% A 10–15 min; 80–0% A 15–25 min; 0% A 25–28 min. The detection wavelength of DAD was set at 230, 260, and 320 nm, and elution was performed at a flow rate of 0.8 mL min$^{-1}$. The flavonoid standard kaempferol was purchased from Sigma–Aldrich Co. Quantification was performed according to the external standard quantification method. Results were expressed as mg per g of dry extract (mg g$^{-1}$ dw).

### 2.4. De Novo Shoot Regeneration from Different Explants of Rocket Seedlings

Cotyledon, hypocotyl, and root explants were cut from five-day-old seedlings and placed in Petri dishes containing 25 mL of basal MS medium with different plant growth regulators, including 2,4-D (2,4–dichlorophenoxyacetic acid, Sigma–Aldrich, Chemie GmbH, Taufkirchen, Germany) at 0.5–2.0 mg $L^{-1}$, BA or TDZ (thidiazuron, Duchefa, The Netherlands) at 1.0 mg $L^{-1}$ + NAA ($\alpha$-naphthaleneacetic acid, Sigma–Aldrich, Chemie GmbH, Taufkirchen, Germany) at 0.1 mg $L^{-1}$, and BA or TDZ at 1.0 mg $L^{-1}$ + NAA 0.1 mg $L^{-1}$ + 5.0 mg $L^{-1}$ AgNO$_3$. Explants were cultivated for shoot induction for seven weeks.

### 2.5. Somatic Embryogenesis from Immature Zygotic Embryos

In order to produce zygotic embryo–donor plants, seeds of rocket variety Domaća rukola were sown into plastic pots in a greenhouse of the Institute of Biological Research "Siniša Stanković" in Belgrade, Serbia (117 m altitude, latitude 44°48′14.44″ N, longitude 20°27′54–47″ E) in the middle of April 2021. Immature siliques were harvested in June 2021. Immature zygotic embryos (1.0–1.5 mm) were isolated from surface-sterilized immature seeds under a stereomicroscope and placed into Petri dishes filed with 25 mL of basal MS medium containing 2,4-D at 0.5, 1.0, and 2.0 mg $L^{-1}$. Cultures were cultivated for three weeks for somatic embryogenesis induction. Somatic embryos were germinated on MS medium with either 30 g $L^{-1}$ PEG (Polyethylene glycol 6000 Fluka), 0.5 mg $L^{-1}$ BA, 0.1–0.5 mg $L^{-1}$ KIN, or plant growth regulator (PGR)–free medium.

### 2.6. Histological Analysis

Samples for histological analysis of somatic embryogenesis were fixed in cold FAA fixative (5 mL 40% formaldehyde: 5 mL acetic acid: 90 mL 70% ethanol) for 30 days at 4 °C, dehydrated with ethanol series, cleared with xylol, and embedded in Paraplast at 58 °C. The samples were sectioned into 8–10 μm thick segments with a microtome (HistoCoreBiocut, Leica Biosystems, Richmond, IL, USA), stained with Delafield haematoxylin, mounted in Canada balsam, and viewed and photographed under a light Leitz DMRB photomicroscope (Leica, Wetzlar, Germany).

### 2.7. Statistical Analysis

All cultures were placed in a completely randomized design. The treatments were repeated at least 2–5 times. For all numerical parameters, ANOVA (analysis of variance) was performed. The difference between means of kaempferol content in cytokinin-treated and control seed-derived plants was evaluated by Student's *t*-test at $p \leq 0.05$. For the analysis of the data on the de novo shoot regeneration and kaempferol content in the cytokinin-treated in vitro shoots, two-way ANOVA was performed. The obtained means were compared by Fisher's least significant difference (LSD) post-hoc test at a 95% confidence level using the StatGraphics Plus for Windows 2.1 (Statistical Graphics Corp., Rockville, MD, USA) software package.

## 3. Results

The results report the establishment of sustainable in vitro culture of the rocket using three different pathways: 1—shoot multiplication from isolated epicotyls of sterile germinated seedlings; 2—de novo shoot regeneration from cotyledons, hypocotyls, and roots of sterile germinated seedlings; and 3—regeneration of somatic embryos from immature zygotic embryos. A schematic representation of these three propagation pathways is shown in Figure 1.

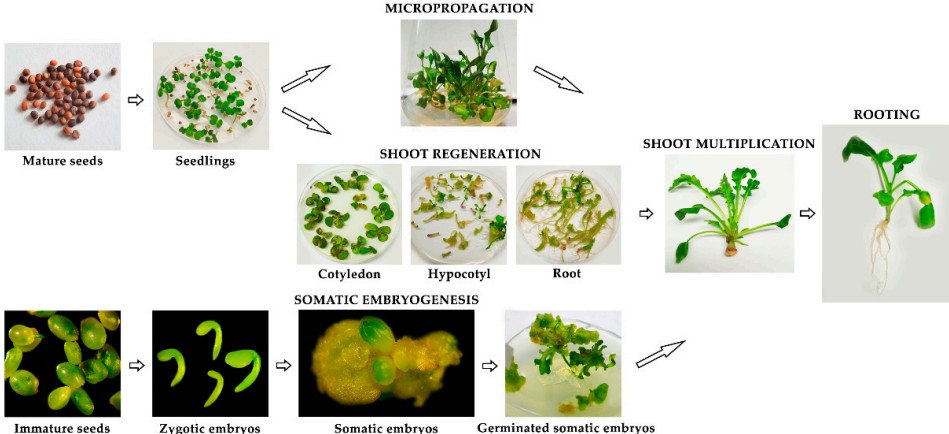

**Figure 1.** Schematic representation of three different pathways for in vitro propagation of the rocket variety Domaća rukola.

### 3.1. Shoot Multiplication from Epicotyl Explants of Seedlings

The procedure used for mature seed disinfection provided 100% of uninfected seeds that germinated at a high rate (about 95%) when placed on a PGR-free medium. The preliminary experiment indicated that the simplest and fastest way to establish an in vitro shoot culture of rocket is to place the epicotyls of germinating seedlings on a medium with cytokinins BA or KIN. The low concentrations (0.1–0.2 mg L$^{-1}$) were more suitable for the growth and development of shoots compared to the higher ones (0.5–2.0 mg L$^{-1}$), which favored the appearance of shoot aberration and hyperhydricity. It was also found that more frequent subculturing of rocket shoots on a fresh medium reduced hyperhydration, so the time between two subcultures was shortened from four to five weeks to three weeks. ANOVA indicated that cytokinin treatment significantly affected the shoot multiplication rate as well as the frequency of hyperhydrated shoots. BA and KIN at 0.2 mg L$^{-1}$ promoted shoot proliferation with multiplication index reaching 4.10 and 3.15, respectively (Table 1, Figure 2a,b). Cytokinins also caused a higher incidence of unviable and hyperhydrated shoots, leading to the shoots yellowing and even necrosis, especially in those multiplied on BA 0.2 mg L$^{-1}$ (Figure 2a).

**Table 1.** Effect of BA and KIN at 0.1 and 0.2 mg L$^{-1}$ on multiplication index and shoot viability of rocket variety Domaća rukola after three weeks of cultivation.

| Cytokinin (mg L$^{-1}$) | Explant No. | Multiplication Index [1] | Viable Shoots (%) | Hyperhydrated Shoots (%) | Necrosis (%) |
|---|---|---|---|---|---|
| BA 0.1 | 279 | 3.45 ± 0.11 [b] | 91.28 | 8.72 | 0.00 |
| BA 0.2 | 100 | 4.10 ± 0.16 [c] | 58.54 | 26.58 | 14.88 |
| KIN 0.1 | 152 | 2.98 ± 0.13 [a] | 90.73 | 9.27 | 0.00 |
| KIN 0.2 | 123 | 3.15 ± 0.14 [ab] | 75.45 | 24.55 | 0.00 |

[1] Multiplication index—(one shoot explant + number of newly formed axillary shoots)/total number of shoot explants per treatment. Data represent mean values of three replications. Means ± SE within the column denoted by different letters are significantly different ($p \leq 0.05$) according to the LSD test.

Although the multiplication was significantly lower when BA and KIN at 0.1 mg L$^{-1}$ were applied (index of multiplication 3.45 and 2.98, respectively), the percentage of viable shoots was the highest (above 90%), with low hyperhidration rate (below 10%) and without necrosis (Table 1, Figure 2c,d). KIN 0.1 mg L$^{-1}$ also induced spontaneous rooting (Figure 2d). BA at 0.1 mg L$^{-1}$ was optimal for rocket shoot multiplication with an index of multiplication of 3.45, providing the formation of clusters with vital and healthy-looking intense green-colored shoots (Figure 2c,e).

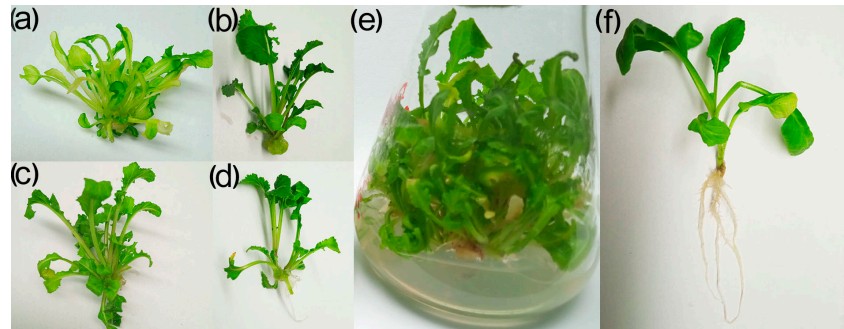

**Figure 2.** In vitro shoot multiplication and rooting of rocket variety Domaća rukola. Multiplied shoot clusters after three weeks of cultivation of epicotyl explants on MS medium with 0.2 mg L$^{-1}$ BA (**a**), 0.2 mg L$^{-1}$ KIN (**b**), 0.1 mg L$^{-1}$ BA (**c**), 0.1 mg L$^{-1}$ KIN (**d**), multiplied shoots after three weeks of cultivation on MS medium containing 0.1 mg L$^{-1}$ BA (**e**). Rooted shoot on ½MS medium supplemented with 0.5 mg L$^{-1}$ IBA for three weeks (**f**).

*3.2. The Content of Secondary Metabolites in Multiplied Shoots Grown In Vitro*

HPLC analysis of methanol extracts of *E. sativa* shoots revealed the presence of two main groups of secondary metabolites. These compounds were tentatively identified as glucosinolates and flavonoid glucosides based on retention times and characteristic UV spectral data in the previously published literature [31] (Figure 3).

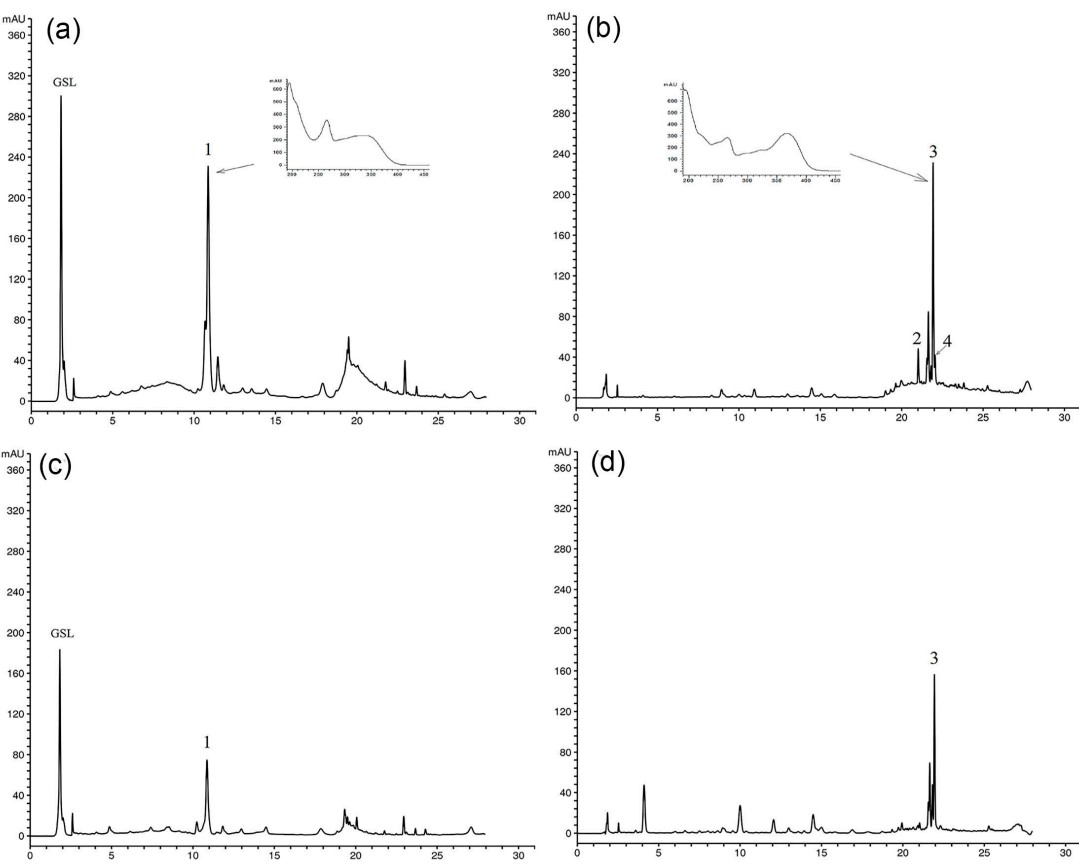

**Figure 3.** HPLC profile (λ = 320 nm) with characteristic UV spectra of dominant compounds of rocket variety Domaća rukola extracts: extract of control plant grown in a greenhouse (**a**) before and (**b**) after acid hydrolysis; extract of in vitro cultured shoots (KIN 0.2 mg L$^{-1}$) (**c**) before and (**d**) after acid hydrolysis. Peak 1—kaempferoldiglucoside; peak 2—quercetin; peak 3—kaempferol; peak 4—isorhamnetin; peak GSL—glucosinolates.

It is evident from the chromatograms that the extracts of the control plants culti­vated in a greenhouse and the plants cultivated in vitro did not differ in their qualitative composition. However, they differed in quantitative content, where the control plants contained higher amounts of secondary metabolites than in vitro-cultured and cytokinin-treated shoots.

To identify and quantify individual flavonoids, their glucosides were hydrolyzed to obtain aglycones since the determination of flavonoid glucosides in plant extracts is difficult due to their large number and structural similarity. The results obtained after acid hydrolysis of *E. sativa* extracts showed that kaempferol was the predominant flavonoid in all samples studied (Figure 3a–d). The highest amount of kaempferol was found in extract from control plants (2.47 mg g$^{-1}$ dw), while in shoots multiplied in vitro, its content was significantly lower (Table 2). According to ANOVA, kaempferol content in in vitro shoots depended on cytokinin type and concentration and their interaction and varied from 0.68 to 1.74 mg g$^{-1}$ dw (Table 2). The highest value of kaempferol was recorded in shoots grown on a medium containing KIN 0.2 mg L$^{-1}$ (Table 2).

**Table 2.** Kaempferol content in methanol extracts of control plants grown in a greenhouse and in vitro–multiplied shoots of rocket variety Domaća rukola after acid hydrolysis.

| Extract Sample | Multiplication Treatment | | Kaempferol Content $\pm$ SE |
|---|---|---|---|
| | **Cytokinin** | **Concentration (mg L$^{-1}$)** | **(mg g$^{-1}$ dw)** |
| In vitro-grown shoots | KIN | 0.1 | 0.68 $\pm$ 0.007 [*a] |
| | | 0.2 | 1.74 $\pm$ 0.010 [*d] |
| | BA | 0.1 | 1.50 $\pm$ 0.010 [*c] |
| | | 0.2 | 1.10 $\pm$ 0.009 [*b] |
| Control | - | - | 2.47 $\pm$ 0.011 |

| ANOVA Source of Variation | Df | Mean Square | *F*-Ratio | *p*-Value |
|---|---|---|---|---|
| Cytokinin | 1 | 0.02484 | 103.730 | 0.000000 |
| Concentration | 1 | 0.31493 | 1314.94 | 0.000007 |
| Cytokinin $\times$ concentration | 1 | 1.59432 | 6656.88 | 0.000000 |

Data represent mean values of three measurements $\pm$SE. Cytokinin treatments denoted by different letters are significantly different according to the LSD test at $p \leq 0.05$ following two-way ANOVA with cytokinin type and concentration as factors. Student's *t*-test was performed to compare differences in kaempferol content in cytokinin-treated in vitro-grown shoots versus seed-derived control, with asterisk (*) denoting significant difference at $p \leq 0.05$.

Doubled KIN concentration (0.2 mg L$^{-1}$) significantly enhanced kaempferol content (2.5-fold) compared to shoots multiplied on lower KIN concentration, while this effect was not found when the same concentration of BA was used. In addition, an even lesser amount (1.10 mg g$^{-1}$ dw) of kaempferol was recorded on BA 0.2 mg L$^{-1}$ compared to plants multiplied in lower concentration (1.50 mg g$^{-1}$ dw) (Table 2).

### 3.3. Rooting of Multiplied Shoots

At least 51.84% of individual shoots, about 1.5–3.0 cm long, have been rooted after cultivation in a medium with $\frac{1}{2}$MS mineral salts for three weeks (Table 3). The addition of IBA at a concentration of 0.1–1.0 mg L$^{-1}$ further promoted the rooting of the shoots. However, it did not significantly affect the number of roots per rooted plant and the length of the longest root (Table 3). The highest rooting (89.38%) was achieved on IBA at 0.5 mg L$^{-1}$, where well-elongated and viable shoots were developed (Table 3, Figure 2f).

**Table 3.** Rooting of multiplied shoots of rocket variety Domaća rukola after three weeks of cultivation on media with increasing concentrations of IBA (0–1.0 mg L$^{-1}$).

| IBA (mg L$^{-1}$) | Explants No. | Rooting (%) | Mean No. of Roots | Mean Length of the Longest Root (mm) |
|---|---|---|---|---|
| 0 | 34 | 51.84 ± 12.61 [a] | 5.12 ± 0.60 [a] | 59.0 ± 11.4 [a] |
| 0.1 | 30 | 73.33 ± 7.70 [ab] | 5.19 ± 0.49 [a] | 52.6 ± 6.7 [a] |
| 0.2 | 36 | 69.52 ± 9.35 [ab] | 5.19 ± 0.36 [a] | 68.4 ± 9.5 [a] |
| 0.5 | 36 | 89.38 ± 4.38 [b] | 4.56 ± 0.25 [a] | 54.0 ± 7.4 [a] |
| 1.0 | 36 | 76.19 ± 9.52 [ab] | 5.29 ± 0.32 [a] | 69.2 ± 8.1 [a] |

Data represent mean values of three repetitions. Means ± SE within each column followed by the same letters are not significantly different according to Fisher's LSD test ($p \leq 0.05$).

### 3.4. De Novo Shoot Organogenesis

Cotyledon, hypocotyl, and root explants cultured on a PGR–free medium for three weeks did not show any signs of shoot regeneration. In order to induce shoot regeneration, different regeneration treatments were applied, including individual 2,4-D at 0.5–2.0 mg L$^{-1}$, BA, and TDZ in combination with NAA, all at a concentration of 0.1 mg L$^{-1}$, with or without AgNO$_3$ at 5.0 mg L$^{-1}$ (Table 4).

The results showed that 2,4-D in all applied concentrations stimulated only the induction of calli at the sites of the intersection of cotyledon, hypocotyl, and root explants, while no shoot regeneration occurred (Table 4). The application of BA+NAA or TDZ+NAA combinations induced DNSO in hypocotyl and root explants, and the frequency of regeneration depended on the PGR combination used and the addition of 5.0 mg L$^{-1}$ AgNO$_3$ (Table 4). However, cotyledons did not show potential for shoot regeneration on any of these formulations tested due to rapid necrosis observed on both types of treatments. The addition of AgNO$_3$ promoted extreme growth and greening of cotyledon explants alongside sporadically developed arachnoid roots, but shoot induction decidedly failed (Table 4).

**Table 4.** Effect of different plant growth regulators on de novo shoot regeneration via organogenesis from cotyledon, hypocotyl and root explants of rocket variety Domaća rukola after seven weeks of cultivation.

| Explant Type | Treatment (mg L$^{-1}$) | Explant No. | Mean No. of Explants with Shoots | Regeneration Frequency (%) | Mean No. of Regenerated Shoots per Regenerative Explant |
|---|---|---|---|---|---|
| | 2,4-D 0.5 | 94 | 0 | 0.00 | 0 [a] |
| | 2,4-D 1.0 | 88 | 0 | 0.00 | 0 [a] |
| | 2,4-D 2.0 | 93 | 0 | 0.00 | 0 [a] |
| cotyledon | BA 1.0+NAA 0.1 | 166 | 0 | 0.00 | 0 [a] |
| | BA 1.0+NAA 0.1+AgNO$_3$ 5.0 | 162 | 0 | 0.00 | 0 [a] |
| | TDZ 1.0+NAA 0.1 | 75 | 0 | 0.00 | 0 [a] |
| | TDZ 1.0+NAA 0.1+AgNO$_3$ 5.0 | 73 | 0 | 0.00 | 0 [a] |
| | 2,4-D 0.5 | 43 | 0 | 0.00 | 0 [a] |
| | 2,4-D 1.0 | 43 | 0 | 0.00 | 0 [a] |
| | 2,4-D 2.0 | 42 | 0 | 0.00 | 0 [a] |
| hypocotyl | BA 1.0+NAA 0.1 | 272 | 25 | 9.19 | 1.68 ± 0.14 [bc] |
| | BA 1.0+NAA 0.1+AgNO$_3$ 5.0 | 166 | 13 | 7.83 | 2.08 ± 0.26 [cd] |
| | TDZ 1.0+NAA 0.1 | 221 | 14 | 6.33 | 1.5 ± 0.17 [b] |
| | TDZ 1.0+NAA 0.1+AgNO$_3$ 5.0 | 130 | 33 | 25.38 | 2.18 ± 0.16 [d] |

**Table 4.** *Cont.*

| Explant Type | Treatment (mg L$^{-1}$) | Explant No. | Mean No. of Explants with Shoots | Regeneration Frequency (%) | Mean No. of Regenerated Shoots per Regenerative Explant |
|---|---|---|---|---|---|
| | 2,4-D 0.5 | 21 | 0 | 0.00 | 0 [a] |
| | 2,4-D 1.0 | 22 | 0 | 0.00 | 0 [a] |
| | 2,4-D 2.0 | 22 | 0 | 0.00 | 0 [a] |
| root | BA 1.0+NAA 0.1 | 78 | 3 | 3.85 | 1.33 ± 0.33 [bcd] |
| | BA 1.0+NAA 0.1+AgNO$_3$ 5.0 | 81 | 14 | 17.28 | 1.57 ± 0.17 [bc] |
| | TDZ 1.0+NAA 0.1 | 72 | 5 | 6.94 | 2.0 ± 0.32 [bcd] |
| | TDZ 1.0+NAA 0.1+AgNO$_3$ 5.0 | 62 | 13 | 20.97 | 2.08 ± 0.24 [cd] |
| **ANOVA Source of Variation** | **Df** | **Mean Square** | **F-Ratio** | **p-Value** | |
| Explant | 2 | 8.67346 | 16.67265 | 0.000000 | |
| Tretman | 6 | 5.06394 | 9.73422 | 0.000000 | |
| Explant × Treatment | 12 | 1.28534 | 2.47076 | 0.005903 | |

The regeneration frequency was calculated as number of shoot-regenerating explants/total number of explants × 100. Data represent mean values of two to five replications, with 15–60 explants per treatment (except root explants on 2,4-D). Means ± SE within the column followed by different letters are significantly different according to Fisher's LSD test ($p \leq 0.05$).

Hypocotyl and root explants were suitable for shoot regeneration in rocket, while the frequency of regeneration depended on the PGR combination used and the addition of AgNO$_3$ (Table 4).

On all media containing BA or TDZ+NAA, callus proliferation occurred after only one to two weeks of culture, both on the epidermis (Figure 4a) and at the intersection sites in the areas of the central cylinder (Figure 4b). The callus was globular, green, or pale yellowish with reddish–purple cells, often with hairs. Induction of regenerating shoots was first noticed after three weeks of cultivation (Figure 4c). According to ANOVA, the mean number of shoots per explant was significantly affected by both explant type, PGR combination, and by their interaction as well (Table 4). After seven weeks of culture, combination BA 1.0+NAA 0.1 mg L$^{-1}$ induced shoot regeneration in 9.19% of hypocotyl explants, while the regeneration response of root explants cultured on the same medium was lower, only 3.85% (Table 4). Regeneration frequency remained less than 10% when TDZ 1.0+NAA 0.1 mg L$^{-1}$ was applied to both explant types (Table 4). However, the addition of AgNO$_3$ into regenerating medium significantly enhanced shoot regeneration in almost all explant type/plant growth regulator combinations. AgNO$_3$ promoted shoot regeneration frequency in root explants cultivated on both BA 1.0+NAA 0.1 mg L$^{-1}$ and TDZ 1.0+NAA 0.1 mg L$^{-1}$ to 17.28% and 20.97%, respectively (Table 4). Unfortunately, many of the shoots regenerated from root explants displayed malformation due to hyperhydration, so they were practically unusable for establishing of permanent shoot culture of rocket.

In hypocotyl explants cultured on TDZ+NAA, the regeneration response reached 25.38% with the addition of AgNO$_3$ compared to 6.33% on the same medium without AgNO$_3$ (Table 4).

Since the healthy-looking shoots were obtained in hypocotyl explants using the combination TDZ+NAA+AgNO$_3$, with satisfactory regenerating frequency and the mean number of shoots per explant (Figure 4d), this protocol can be selected as optimal for shoot regeneration of rocket. The regenerated shoots were excised and further multiplied on basal MS medium with BA 0.1 mg L$^{-1}$, and rooted on 1/2MS medium with 0.5 mg L$^{-1}$ IBA, as was used previously.

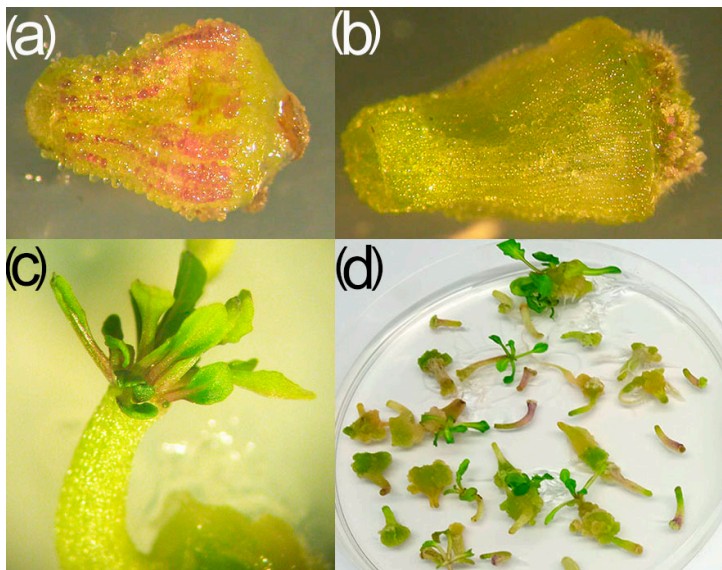

**Figure 4.** De novo shoot regeneration via organogenesis from hypocotyl explants of rocket variety Domaća rukola cultivated on MS medium supplemented with TDZ 1.0+NAA 0.1+AgNO$_3$ 5.0 mg L$^{-1}$. Small callus formation started after two weeks of cultivation on hypocotyl epidermis (**a**) and on the intersection sites in the area of central cylinder (**b**); beginning of shoot regeneration after three weeks of cultivation (**c**); hypocotyl explants with viable regenerated shoots (**d**).

### 3.5. Somatic Embryogenesis from Immature Zygotic Embryos

For somatic embryogenesis induction, zygotic embryos (1.0–1.5 mm) isolated from immature siliques of rocket plants (Figure 5a,b) were cultivated on MS medium containing 2,4-D at 0, 0.5, 1.0, and 2.0 mg L$^{-1}$. Immature zygotic embryos cultured on basal MS medium without 2,4-D germinated into whole plants, and no somatic embryogenesis was observed, while those cultured on media with 2,4-D began to thicken and form embryogenic calli after seven days of cultivation (Figure 5c). First SEs were observed after seven to ten days of cultivation. Most of them arose from a previously formed small portion of the callus (Figure 5d), although there was a direct formation of somatic embryos as well (Figure 5e,f). The regeneration process was asynchronous, so SEs in different stages of development could be seen on one explant at the same time (Figure 5e,f). The embryos were small, properly developed, viable, and mainly green-colored.

Generally, no differences in morphological characteristics and phenotype were observed between SEs regenerated on media with different concentrations of 2,4-D. A large portion of individual SEs was easily separated from the initial explant. However, somatic embryos fused with maternal tissue have also been observed. These structures were generally extremely rigid and inflexible, and most of them were found on 2.0 mg L$^{-1}$ 2,4-D-containing medium.

The efficiency of embryogenesis depended on the concentration of 2,4-D used (Table 5). Satisfactory embryogenic efficiency was achieved at the lowest concentration of 2,4-D (0.5 mg L$^{-1}$), with a minimal adverse effect on immature embryos as well, since only 0.71% of them were necrosed, along with 4.25% of non-responsive explants. Necrosis of explants increased with increasing concentration of 2,4-D. So, the highest necrosis was observed in immature embryos cultured in medium with the highest 2,4-D at 2.0 mg L$^{-1}$, indicating that this concentration is less favorable (Table 5).

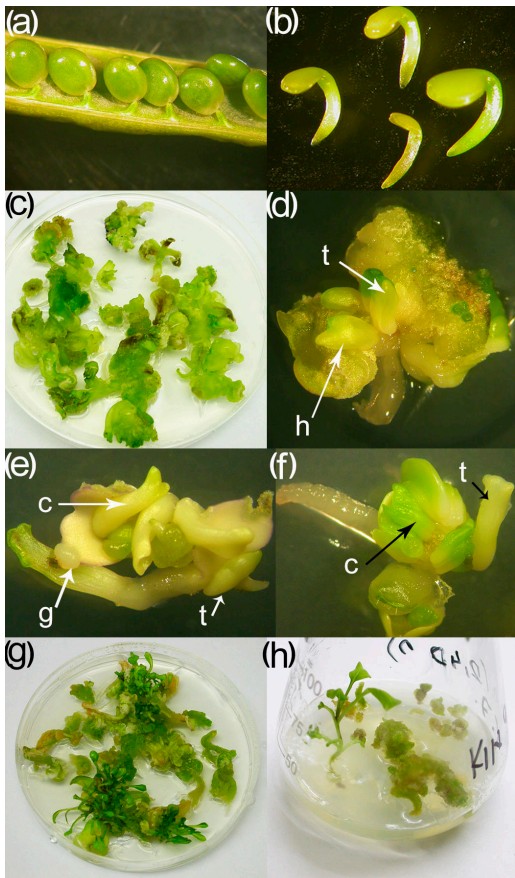

**Figure 5.** Induction of somatic embryogenesis from immature zygotic embryos of rocket variety Domaća rukola by treatment with 1.0 mg L$^{-1}$ 2,4-D. Silique with immature seeds (**a**); immature zygotic embryos (**b**); somatic embryos (SEs) formation (**c**); embryogenic callus with somatic embryos in different stages of development: note heart (h-arrow) and torpedo (t-arrow) (**d**); asynchronous somatic embryos development from embryogenic callus formed on immature zygotic embryos: note globular (g-arrow), torpedo (t-arrow), and cotyledonary (c-arrow) stage (**e**); direct somatic embryogenesis induced on immature zygotic embryos by 1.0 mg L$^{-1}$ 2,4-D: note torpedo (t-arrow) and cotyledonary (c-arrow) stage (**f**); SEs germination after three weeks of cultivation on MS medium with 0.5 mg L$^{-1}$ KIN (**g**); individual SEs germinated into plants (**h**).

**Table 5.** Effect of 2,4-D on somatic embryogenesis from immature zygotic embryos of rocket variety Domaća rukola after three weeks of culture.

| 2,4-D (mg L$^{-1}$) | No. of Explants | Explants with SE (%) | Mean No. Somatic Embryos per Embryogenic Explant | Callus (%) | Necrotic Explants (%) | Explants without Response (%) |
|---|---|---|---|---|---|---|
| 0.5 | 335 | 71.48 ± 4.29 [a] | 4.25 ± 0.27 [a] | 23.39 ± 4.01 [a] | 0.71 ± 0.51 [a] | 4.25 ± 0.60 [b] |
| 1.0 | 226 | 76.64 ± 5.30 [a] | 5.13 ± 0.25 [b] | 11.42 ± 4.42 [a] | 9.38 ± 0.27 [b] | 2. 56 ± 2.59 [ab] |
| 2.0 | 213 | 64.54 ± 6.20 [a] | 4.00 ± 0.36 [a] | 24.14 ± 4.29 [a] | 11.32 ± 0.27 [b] | 0.00 ± 0.00 [a] |

Data represent mean values of two to three replications. Means ± SE within the column followed by different letters are significantly different according to Fisher's LSD test ($p \leq 0.05$).

Immature zygotic embryos cultivated in a medium containing 1.0 mg L$^{-1}$ 2,4-D showed the highest but statistically insignificant embryogenic response (76.64%) with the highest number of regenerated SEs per regenerative explant (5.13). The incidence of callus was the lowest, while necrosis was observed in less than 10% of isolated explants (Table 5). Therefore, this medium was chosen as optimal for the production of somatic embryos from immature zygotic embryos of rocket variety Domaća rukola.

To germinate into plants, single SEs were transferred to basal MS medium and MS medium supplemented with 30 g L$^{-1}$ PEG 6000 or with BA or KIN at 0.1 and 0.5 mg L$^{-1}$. Germination of somatic embryos was observed only on media containing cytokinins, indicating their necessity for further development of SEs. Most somatic embryos cultivated on cytokinin-free medium became necrotic and perished over time (Table 6). KIN 0.5 mg L$^{-1}$ was the most suitable for plant development, providing the highest embryo germination (almost 23 of 30) that was later developed into vital plantlets without hyperhydration (Table 6, Figure 5g,h).

**Table 6.** Germination of rocket variety Domaća rukola somatic embryos (SEs) into plants after three weeks of cultivation on different maturation MS media.

| Treatment | No. of SEs Established | Mean No. of Germinated SEs | Hyperhydrated SEs (%) |
|---|---|---|---|
| PEG 6000 30 g L$^{-1}$ | 35 | $0.0 \pm 0.0$ | - |
| PGR-free | 35 | $0.0 \pm 0.0$ | - |
| BAP 0.5 mg L$^{-1}$ | 35 | $6.70 \pm 1.6$ | 80 |
| KIN 0.1 mg L$^{-1}$ | 35 | $12.20 \pm 1.9$ | 0.0 |
| KIN 0.5 mg L$^{-1}$ | 30 | $22.50 \pm 1.4$ | 0.0 |

Data represent mean values of three replications ±SE.

*3.6. Histological Analysis of Somatic Embryogenesis*

Direct and indirect somatic embryogenesis was determined by histological analysis in the rocket variety Domaća rukola. Meristematic centers were observed in the embryogenic callus, both on the callus surface and in deep layers. Somatic embryos at all developmental stages were observed in the sections of embryogenic calli (Figure 6a–d). SEs at the globular stage (Figure 6a), either single or in groups, were quite separable from the surrounding callus tissue. These globular structures gave rise to heart-shaped SEs (Figure 6b), which further developed into bipolar embryos at the torpedo stage (Figure 6c). In SEs at the cotyledonary stage, the shoot pole with cotyledons and shoot meristem and the root pole with root meristem beneath the root cap could be distinguished (Figure 6d).

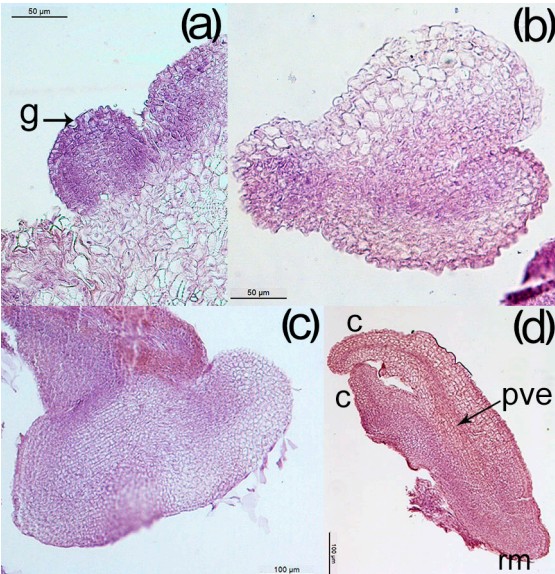

**Figure 6.** Development of somatic embryos from callus induced from immature zygotic embryos of rocket variety Domaća rukola cultured on MS medium with 1.0 mg L$^{-1}$ 2,4-D. Globular, g, (**a**), and heart stage of somatic embryo (**b**) (bar = 50 µm). Torpedo stage (bar = 100 µm) (**c**). Cotyledonary stage (**d**): note discernible root pole with root meristem (rm), cotyledons (**c**), and provascular elements (pve) (bar = 100 µm).

## 4. Discussion

Rocket variety Domaća rukola is recognized as a potent cultivar for its beneficial nutritional and health properties. However, it is very sensitive to pests, especially aphides and whiteflies, which compromise the immunity of the plants and make them more susceptible to other pests and diseases [17,32]. Tissue culture could be used to improve some plant characteristics, but its application is also characterized by certain limitations, such as complications and the high cultivar-dependence efficiency of in vitro procedures. This is the reason why a number of research on in vitro regeneration protocols have been developed via both de novo shoot organogenesis and somatic embryogenesis using different combinations of plant growth regulators, mostly BA, KIN, and 2,4-D in combination with NAA or IAA [16,21,22,27,33].

To establish in vitro shoot culture of rocket variety Domaća rukola, epicotyls of in vitro-germinated seedlings were cultivated on MS media containing low concentrations of BA or KIN that were previously shown suitable for rocket micropropagation or shoot regeneration [24,26,33]. Although the higher concentration of both BA and KIN was more efficient for shoot multiplication, according to the shoot multiplication index, BA at 0.1 mg L$^{-1}$ was shown to be a better choice for multiplication due to the production of vital and healthy-looking shoots, which were further successfully rooted in $\frac{1}{2}$MS medium with 0.5 mg L$^{-1}$ IBA. HPLC analysis confirmed the same qualitative content of flavonoid compounds in control greenhouse-grown plants and in vitro multiplied shoots, with kaempferol as a predominant flavonoid; however, their content was significantly lower in in vitro cultured shoots. Most often, the content of desired secondary metabolite in in vitro-grown plants can initially be quite low, so screening for high-yield and fast-growing lines, as well as appropriate culture conditions, should be undertaken [34]. It is interesting to note that shoots multiplied at KIN 0.2 mg L$^{-1}$ contained a 2.5-fold higher amount of kaempferol compared to those multiplied at KIN 0.1 mg L$^{-1}$. Grzegorczyk-Karolak et al. [35] reported that exogenously supplied cytokinins could significantly affect the production of polyphenolic metabolites in shoots of *Scutellaria alpina*, where the urea-type cytokinin TDZ was more effective than BA, kinetin, and zeatin in inducing the production of baicalin, wogonoside, and verbascoside. Exogenously applied KIN was found to increase growth performance and metabolite production in plants of *Solanum lycopersicum* under NaCl stress by upregulating antioxidant metabolism and osmolyte accumulation [36]. Nowadays, Khan et al. [37] reported that nanotized KIN improved the yield of essential oil and active constituents in mint plants by enhancement of their physiological attributes. Overall, these observations indicate the suitability of KIN application for exploiting the genetic potential of plants for growth and metabolite production. In an attempt to regenerate shoots from different explants of rocket variety, Domaća rukola, as a starting procedure for potential genetic transformation, various growth regulators, including 2,4-D, cytokinins BA, and TDZ in combination with NAA, were used. No regenerative response of explants cultivated in a medium containing 2,4-D as well as low regeneration potential of hypocotyl and root explants cultivated on BA or TDZ combined with NAA, undoubtedly confirmed the recalcitrance of this rocket variety for in vitro shoot regeneration. In other species and cultivars of rocket, large differences in the regeneration response were also found. Sikdar et al. [27] regenerated plantlets of the wild rocket from mesophyll protoplasts via organogenesis with a frequency of 15.70%, using 2,4-D, NAA, BA, and GA$_3$. Leskovšek et al. [38] regenerated shoots from isolated microspores of rocket in a medium with activated charcoal and pointed out the effect of genotype, both among seed lots and among individual plants, as a key factor affecting regeneration. On the other hand, Abbasi et al. [26] increased the frequency of organogenesis response in leaf explants that produced calli to about 80–90%, using MS media supplemented with higher concentrations of cytokinin, including 2.0 and 5.0 mg L$^{-1}$ of BA and 5 and 10 mg L$^{-1}$ KIN in combination with 1.0 mg L$^{-1}$ NAA. Unfortunately, increasing concentrations of BA and TDZ to 2.0 and 4.0 mg L$^{-1}$ did not promote regeneration response in the variety Domaća rukola. However, a considerable increase in the frequency of regeneration of hypocotyl explants (from 6.33% to 25.38%) and root explants

(from 6.94% to 20.97%) was achieved by the application of AgNO$_3$ to media containing TDZ 1.0+NAA 0.1 mg L$^{-1}$. The effect of AgNO$_3$ on promoting shoot and root growth has previously been reported in various plant species. Venkatachalam et al. [39] indicated that AgNO$_3$ improved shoot regeneration in *Prosopis cineraria*. Bhadane et al. [40] also showed the positive role of AgNO$_3$ in combination with BA and KIN in the induction of multiple shoots in *Carissa carandas*. AgNO$_3$ increased the frequency of *Vigna mugo* shoot regeneration as well [41]. The addition of AgNO$_3$ to BA-supplemented MS medium not only improved the bud regeneration response but also the bud elongation in *Alternanthera sessilis* [42] and enhanced polyamines to improve shoot regeneration in potato [43]. In addition, in Chinese cabbage, which is recalcitrant to genetic transformation, an effective method for genetic transformation mediated by *Agrobacterium* was established by adjusting the concentration of hormones and AgNO$_3$ in co-cultivation and selection media [44]. AgNO$_3$ promoted the growth of shoots and roots in tissue culture by inhibiting the activity of ethylene through the Ag 2+ ions, that is, by reducing the capacity of receptors to bind ethylene [45,46].

However, apart from the modest regeneration potential of rocket seedling explants that generally does not excide 26%, other potential problems with the use of shoot organogenesis may include complicated sequential changes of culture media and possible occurrence of somaclonal and chimeric variants among regenerated and transformed plants. Somatic embryogenesis can be an efficient alternative for in vitro regeneration. Immature zygotic embryos of rocket variety Domaća rukola were shown to be the best option for in vitro regeneration with almost 80% frequency of somatic embryogenesis, and about five somatic embryos regenerated per responsive explant when treated with 1.0 mg L$^{-1}$ 2,4-D. In most plant species, embryo-competent cells in the explant tissue generally require certain stimuli to develop into somatic embryos. The growth regulator 2,4-D has been successfully used to induce somatic embryogenesis in many plant species, including the Brassicaceae family [47].

Histological analysis confirmed that there were no vascular connections between maternal tissue and developing embryos, which had well-defined root and cotyledon poles and which ultimately germinated into viable plants. Previously, Zhang et al. [16] studied the regeneration potential of cotyledons and hypocotyls of rocket cultivar Qingcheng and found that up to 83.90% of cotyledons and 64.80% of hypocotyls could produce embryogenic calli, with the optimal hormone combination of 4.52 mM 2,4-D. Chen et al. [12] achieved satisfactory somatic embryogenesis efficiency in a rocket ecotype from the arid regions of northwestern China when compared to that achieved in the present study. The protocol involves the use of cotyledons, petioles, and hypocotyls as explants and their cultivation on the media containing 2,4-D in combination with KIN or BA. The combination of hypocotyls and 2,4-D 1.0+KIN 0.3 mg L$^{-1}$ was chosen to be a good system for somatic embryogenesis since as many as 90% of hypocotyl explants produced embryogenic callus, and 48% of somatic embryos germinated into seedlings after they were transferred to PGR-free MS medium. However, the application of 2,4-D on different explants of seedlings of the variety Domaća rukola did not result in any regeneration response. In rocket, immature zygotic embryos have rarely been used for somatic embryo regeneration compared to seedling explants. Ashloowalia [21] cultured zygotic embryos of *E. vesicaria* ssp. *sativa*, taken from immature seeds at various stages of development, in MS medium with 0.5 mg L$^{-1}$ NAA and 5.0 mg L$^{-1}$ BA, and found that only a few produced somatic embryos while organogenesis was induced in the majority of explants. However, immature embryos have been successfully used as optimal explants to establish somatic embryogenesis in other important *Brassica* vegetable species, such as cabbage and cauliflower [48]. A high frequency of direct somatic embryogenesis of specific cabbage and cauliflower lines was obtained by culturing immature zygotic embryo explants on a PGR-free induction medium [48].

Somatic embryo maturation and germination into viable plants have been shown to be a critical step in somatic embryogenesis in many plant species, including rocket. Therefore, various procedures have been carried out to promote this process. Ashloowalia concluded [21] that once somatic embryos of the rocket were produced, growth regulators

were not required for their germination, as those subcultured on media supplemented with 5.0 mg L$^{-1}$ NAA or with 0.1 mg L$^{-1}$ indole acetic acid (IAA) and 1.0 mg L$^{-1}$ zeatin did not develop further and became necrotic. Accordingly, only the transfer of somatic embryos to 1/2MS or MS medium without growth regulators resulted in plant development, but the efficacy was not mentioned. When Chen et al. [12] transferred calli-containing somatic embryos, regenerated from cotyledon explants of rocket, to different mineral nutrition media without hormones, the maturation of somatic embryos varied from 4.30% to 22.90%, with MS medium being the most effective. A higher maturation of somatic embryos of 34% was achieved when the same calli were cultured in MS medium with increasing sucrose content (optimal 60–80 g L$^{-1}$). Zhang et al. [16] described that desiccation treatment improved the germination of somatic embryos after rehydration, resulting in more than 70% of embryos surviving and undergoing further development. Zhang et al. [16] also used different protocols to achieve better maturation of somatic embryos of rocket cv. Qing-Cheng, including different concentrations of abscisic acid (ABA), polyethylene glycol (PEG)-4000, sucrose, and activated charcoal (AC). Although sucrose and AC also enhanced the maturation of somatic embryos, they reported that MS with PEG (45 g L$^{-1}$) was the most favorable medium in which 22.10 embryos were converted into plants from 2.5 g of established SEs, as evidenced by the development of embryos at the cotyledonary stage, a lower percentage of abnormal structures, and enhanced cotyledon development. In the present study, however, the addition of PEG as well as the transfer to MS medium without PGRs, did not positively affect the maturation of somatic embryos of rocket variety Domaća rukola. Namely, it was shown that the inclusion of cytokinin is necessary for the further development of somatic embryos, and the addition of KIN in the amount of 0.5 mg L$^{-1}$ enabled about 70% of somatic embryos to turn into plants. Cytokinins have also been shown to be effective for somatic embryo conversion in other *Brassica* species. Pavlović et al. [48] reported that the use of BA or KIN at a concentration of 1.0 mg L$^{-1}$ promoted the conversion of somatic embryos to plantlets up to 55.50% in cabbage and 78.60% in cauliflower and improved the quality of the regenerated plantlets.

Somatic embryogenesis from immature embryos can be used extensively for breeding purposes to rescue embryos in interspecific hybridization of incompatible genotypes and propagation of hybrid lines [49]. It is also a favorable artificial/synthetic seed generation system and available means of producing disease-free plant material. Compared to shoot organogenesis, genetically modified plants acquired through somatic embryogenesis produce fewer chimeras [50]. In addition, immature zygotic embryos as young tissue are more susceptible to *Agrobacterium* infection, and this may contribute to more efficient genetic transformation.

## 5. Conclusions

In the present study, an efficient protocol was developed for shoot multiplication and regeneration of recalcitrant rocket variety Domaća rukola. BA at 0.1 mg L$^{-1}$ and frequent subculture over a period of three weeks was optimal for shoot multiplication from seedlings of epicotyl with a multiplication index of almost 3 and only 9.30% of hyperhydrated shoots without necrosis. Kaempferol was found to be the predominant flavonoid in both the control and in vitro shoots. AgNO$_3$ significantly improved de novo shoot organogenesis in both hypocotyl and root explants. The combination of TDZ 1.0+NAA 0.1+AgNO$_3$ 5.0 mg L$^{-1}$ and hypocotyl explants was selected as optimal for shoot regeneration, yielding healthy shoots with a satisfactory regeneration rate of 25.38% and an average number of 2.18 shoots per regenerating explant. Somatic embryogenesis from immature zygotic embryos induced by 1.0 mg L$^{-1}$ 2,4-D proved to be the best regeneration system for recalcitrant rocket variety Domaća rukola that offers the highest frequency of regeneration (76.64%), and the number of regenerated somatic embryos per embryogenic explant (5.13), with a low incidence of callus and necrosis. Rocket shoots produced in vitro could be considered a promising source for the production of the flavonoid kaempferol, which has a pronounced antioxidant potential.

**Author Contributions:** Conceptualization, N.B. and B.V.; methodology, B.V., M.S. and D.K.-M.; validation, D.K.-M. and T.M.; formal analysis, M.S.; investigation, M.P., T.Ć. and N.B.; writing—original draft preparation, N.B. and B.V.; writing—review and editing, D.K.-M. and T.M.; visualization, T.Ć. and M.P.; supervision, B.V. All authors have read and agreed to the published version of the manuscript.

**Funding:** This research was funded by the Ministry of Science, Technological Development, and Innovation of the Republic of Serbia under contract 451-03-47/2023-01/200007.

**Data Availability Statement:** The data presented in this study are available from the authors.

**Acknowledgments:** The authors acknowledge their gratitude to the Ministry of Science, Technological Development, and Innovation of the Republic of Serbia for financial support of this research.

**Conflicts of Interest:** The authors declare no conflict of interest.

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
