# Peer review of "In Vitro Shoot Multiplication and Regeneration of the Recalcitrant Rocket (Eruca sativa Mill.) Variety Domaća Rukola"

_horticulturae, doi:10.3390/horticulturae9050533_

Round 1
Reviewer 2 Report
The authors described a protocol for in vitro shoot multiplication and regeneration via organogenesis and somatic embryogenesis of Eruca sativa. The research is new and interesting. However, the manuscript needs to be improved, particularly regarding the lack of quantitative data (and relative statistics) relating to the conversion of immature zygotic embryos into shoots and of a careful discussion and comparison with other works. Moreover, the manuscript requires some modifications and improvements before being accepted.
Major corrections
Results / Discussion
-lanes 476 to …: The authors first cited the manuscript by Zhang et al. (16) and stated that “but little has been described regarding the shoot formation”, and subsequently they discuss other works concerning somatic embryogenesis in the same species. The formation and development of shoots from somatic embryos is one of the delicate and most important phases of the entire shoot regeneration process. Authors reported in the Results paragraph the these results only in two lanes (371 to 372) and in the Discussion paragraph they do not compare their results with those obtained by others. For these reasons, they have to introduce a table with quantitative data related to the conversion zygotic embryos vs. shoots and describe that in the Results paragraph and discuss/compare in detail in the Discussion paragraph.
Minor corrections
Introduction
- lanes 52 to 58: authors have to introduce references to support their statements
- lanes 63 to 64: substitute “… reported organogenic shoot regeneration using …” with ““… reported shoot regeneration vai organogenesis using …”
- lanes 63 to 67: authors have to exclude from the cited references (17-22) the n.17 because it concerns the shoots regeneration via somatic embryogenesis and not by organogenesis
- lane 68: substitute “… to the organogenic regeneration of shoots, the …” with ““… to the shoot regeneration via organogenesis, the …”
- lane 80: substitute “… regenerative response.” with “… regenerative response (data not shown).”
Results
- lane 198: substitute “… shoot multiplication treatment … with “…cytokinin treatment …”
- lane 209: substitute “… was lower when lower BA and KIN … with “… was significantly lower and lower when BA and KIN …”
- lanes 236 and 238: substitute “… was lower and …” with “… was significantly lower and …”, and “… KIN concentration enhanced …” with “… KIN concentration significantly enhanced …” Every time that authors reported in tables and figures statistic data they have to introduce in the manuscript the terms “significant/tly” and/or “not significant/tly”
- lane 255: substitute “… weeks, 50% of them …” with “… weeks, at least 50% of them …”
- lanes 291 to 292: substitute “… and regeneration treatment, and by …” with “… and plant growth regulator combination, and by …”
- lane 297: substitute “… shoot regeneration.” with “… shoot regeneration in almost explants type/plant growth regulator combinations.
- lane 302: substitute “… of rocket.” with “… of rocket (data not shown).”
- lane 371: substitute “… over time.” with “… over time (data not shown).”
Discussion
- lanes 392 to 395: authors have to introduce references to support their statements
- lane 412: substitute “… content was lower in …” with “… content was significantly lower in …”
- lane 433: substitute “… for in vitro regeneration.” with “…for in vitro shoot regeneration.”
- lanes 444 to 445: substitute “However, a significant increase in …” with “However, a considerable increase in …”
Reviewer 3 Report
The submitted communication entitled: "In vitro Shoot Multiplication and Regeneration of Recalcitrant Rocket (Eruca sativa Mill.) Variety “Domaća rukola”", falls within the scope of the journal.
The paper aimed to establish a feasible protocol for shoot regeneration and propagation of a rocket variety called "Domaća rukola" using plant tissue culture. The study found that using Murashige and Skoog (MS) medium with 0.1 mg/L BA and frequent subculture over a period of 3 weeks was optimal for shoot multiplication. Different concentrations of 2,4-D, BA or TDZ in combination with NAA with or without AgNO3 were tested for shoot organogenesis from seedling explants, with the best results achieved using hypocotyl explants cultured on MS with a combination of TDZ 1.0 +NAA 0.1 + AgNO3 5.0 mg/L. The study also found that somatic embryogenesis from immature zygotic embryos was the best way to regenerate the recalcitrant rocket cultivar, with the highest embryogenic efficiency achieved in explants cultured on MS medium containing 1.0 mg/L 2,4-D. Overall, the study indicates that somatic embryogenesis from immature zygotic embryos is the most suitable protocol for recalcitrant rocket transformation and breeding.
Some grammatical errors needed to be fixed before considering the paper for publication.
Reviewer 4 Report
Dear authors
Your manuscript deals with establishment of in vitro protocols for a very specific Eruca sativa variety (population) that you describe as facing very production problems. I suppose that it can be important for you as a, eventual, local, or regional common variety, as the name suggests. As you refer, your research objectives are specific for a specific genotype or genotype pool, of a species that for which there are available a large panoply of varieties (e.g. this is not an internationally important wine variety).
I am sure that you also agree that a large amount of data have been already published (during the last 3 decades) on the topic of your research and that in your article there are not any sound and novel result or methodology that could be of major interest to the readers of an international scientific journal.
Nevertheless, I call also your attention to the need of an exhaustive language revision, elimination of non-supported by real data, apologetic phrases, e.g. referring to breeding, etc. (example: hybridization is a very good plant breeding tool for genetic introgression) and other things as, when first time referred, to precede the acronyms with the full name of the compound, etc.)
With My Best Regards
Reviewer 5 Report
The manuscript describes elaboration of protocols for effective micropropagation of Eruca sativa. This is an interesting study reporting results of very laborious investigations, however some aspects of it requires clarification.
Specific comments:
1) Abbreviations when used for the first time should be explained, further Murashige and Skoog medium (MS, 1962) – the year of publication should be added;
2) Typing errors should be removed.
3) Results:
3.1.: “normal shoots growth than higher ones..” – please clarify what means normal;
“ANOVA indicated that the shoot multiplication treatment significantly affected the frequency of shoot hyperhydration rate and the multiplication index.” – unclear, and should be re-write;
What means 4.1 or 3.15? Further 3.45?
Table 1 – “… on shoot multiplication..” – what? Index? Number? Further it is not clear how multiplication index was calculated, starting form one bud? More?
“Although the index of multiplication was lower when lower BA and KIN at 0.1 mg L-1 were applied (3.45 and 2.98, respectively),” – unclear; re-write;
Figure 3: nothing has been said about other standard compounds than kaempferol before; further it is not clear what means “Treatments denoted by different letters are significantly different..” – each value is denoted with different letter so, accordingly to Authors all means is statistical significant; similarly in Table 3;
Table 4: it is unclear what data is presented – why repeated 3-times with different values?
Figure 4: just Shoot regeneration – this is organogenesis;
“The lowest 2,4-D at 0.5 mg L-1..” – At the lowest concentration of 2,4-D …
4) Conclusion:
The optimal concentration of BA should be given;
The last sentence should be removed – induction of somatic embryos from transformed tissues was not a subject of this study.
Round 2
Reviewer 5 Report
The manuscript has been improved and I can recommend it for publication.
Author Response
Dear reviewer,
Thank you for your efforts to review the manuscript and provide the comments for considerable improvement.
Sincerely,
The authors